# Recent Developments in the Microencapsulation of Fish Oil and Natural Extracts: Procedure, Quality Evaluation and Food Enrichment

**DOI:** 10.3390/foods11203291

**Published:** 2022-10-20

**Authors:** Trinidad Perez-Palacios, Jorge Ruiz-Carrascal, Juan Carlos Solomando, Francisco de-la-Haba, Abraham Pajuelo, Teresa Antequera

**Affiliations:** Meat and Meat Product University Institute (IProCar), University of Extremadura, Avda. de las Ciencias s/n, 10003 Cáceres, Spain

**Keywords:** microencapsulation, fish oil, antioxidants, emulsions, homogenization, food enrichment

## Abstract

Due to the beneficial health effects of omega-3 fatty acids and antioxidants and their limited stability in response to environmental and processing factors, there is an increasing interest in microencapsulating them to improve their stability. However, despite recent developments in the field, no specific review focusing on these topics has been published in the last few years. This work aimed to review the most recent developments in the microencapsulation of fish oil and natural antioxidant compounds. The impact of the wall material and the procedures on the quality of the microencapsulates were preferably evaluated, while their addition to foods has only been studied in a few works. The homogenization technique, the wall–material ratio and the microencapsulation technique were also extensively studied. Microcapsules were mainly analyzed for size, microencapsulation efficiency, morphology and moisture, while in vitro digestion, flowing properties, yield percentage and Fourier transform infrared spectroscopy (FTIR) were used more sparingly. Findings highlighted the importance of optimizing the most influential variables of the microencapsulation procedure. Further studies should focus on extending the range of analytical techniques upon which the optimization of microcapsules is based and on addressing the consequences of the addition of microcapsules to food products.

## 1. Introduction

Microencapsulation technology arose in the 1950s from the development of dye capsules to be incorporated into paper [1]. Currently, microencapsulation can be defined as a set of technologies aiming to protect sensitive compounds from the external environment and further control their release. For that, the labile compounds, constituting what is known as the core, are entrapped by being surrounded by a shell material (the wall). Microencapsulation has been mainly applied in the pharmaceutical industry, followed in decreasing order by food, cosmetic, textile, biomedical, agricultural and electronic sectors [2].

Microencapsulation of bioactive compounds has attracted the attention of food researchers and industries since most of those compounds (e.g., polyunsaturated fatty acids (PUFAs), antioxidants, vitamins and probiotic bacteria) have limited stability on exposure to environmental factors such as heat, oxygen and light. Besides overcoming stability problems, this technique reduces the perception of possible off-flavors or colors from the encapsulated material; facilitates storage; extends shelf life without adverse influence on the physical, chemical or functional properties of food products; and is of easy application in the food industry (typically it simply involves a powder addition during mixing of ingredients), cheap and scalable [3]. Most relevant reviews in the last few years about microencapsulation of bioactive compounds in the food area have focused on a specific bioactive compound or its source, principally anthocyanins [4], phenolic compounds [5], polyphenols [6], carotenoids [7], vitamins [8,9], oils [10,11] and cells [12], or have reviewed some microencapsulation techniques, specifically complex coacervation [13], spray-chilling [14] and liposomal [15] techniques. 

The beneficial effects of omega-3 PUFAs, principally eicosapentaenoic acid (EPA, C20:5 n-3) and docosahexaenoic acid (DHA, C22:6 n-3), are well known and include decreasing the risk of cardiovascular disease and some types of cancer [16] and preventing neurodegenerative and inflammatory diseases [17,18]; however, there is an insufficient consumption of fish, seafood or algae to reach the recommended intake of EPA plus DHA, which is around 0.25 g per person and day [19]. On the other hand, antioxidant compounds constitute the first line of defense mechanism against free radicals, which are involved in various diseases, such as cancer, asthma, metabolic disorders and neurodegenerative disorders. Over the years, natural, semi-synthetic and synthetic free radical scavengers have gained much attention [20]. Nowadays, the focus is on natural antioxidants from by-products of the agro-industry, which are a valuable resource for utilization in health-enhancing foods [21].

In this context, there has been an increasing interest in the microencapsulation of EPA and DHA sources, mainly fish oil, and natural antioxidant compounds. This is reflected in the number of publications on these topics, which has experienced a significant increase in the last few years (Figure 1). However, to the best of the authors’ knowledge, no specific review on these topics has been recently published. Thus, the present work aims to review the most recent developments in the microencapsulation of fish oil and natural antioxidant compounds to fortify foods, focusing on the key objectives of the developments and the quality evaluation of the microcapsules.

## 2. Search Criteria

The literature search was performed through Scopus, Science Direct and Web of Science, with the keywords “microencapsulation” in combination with “fish oil” or “antioxidant”, limiting the areas of interest (Agricultural and Biological Sciences) and time (2017–2022), in October and November 2021. In this way, 35 and 170 documents resulted, respectively. Due to the high number of papers about microencapsulation of antioxidants, the range of time was shortened (2020–2022). Subsequently, the research articles fulfilling the aim of the present review were selected, resulting in 27 and 36 documents, respectively, to be exhaustively analyzed.

These recent publications on microencapsulation of fish oil and natural antioxidants have principally focused on evaluating the constituents, especially the wall material, of emulsions/solutions for microencapsulation, and the procedures for emulsion/solution preparation and microencapsulation, with only a few studies dealing with the addition of microencapsulates for food enrichment. These key aspects have been reviewed in detail and further discussed and explained.

## 3. Constituents of Emulsions/Solutions for Microencapsulation

Before microencapsulation, a stable emulsion/solution must be prepared, and its constituents strongly influence the quality of obtained microcapsules. Figure 2 depicts the extent of use of different polymers for microencapsulation of fish oil (Figure 2A) and antioxidants (Figure 2B), with three main different types: saccharides, which were the most frequently used, followed by animal and vegetable proteins. The extent of use of saccharides has been higher for microencapsulated antioxidants in comparison to fish oil microencapsulations (77.8 vs. 52.6%, respectively), while animal proteins have a higher degree of use for fish oil than for antioxidant microencapsulation (34.2% vs. 11.1%), and both types of microcapsules showed a similar extent of use of vegetable proteins (13.16% and 11.11, respectively). Whey protein (isolate (WPI) or concentrate (WPC)) and maltodextrin have been predominantly used in most recent studies on fish oil microcapsules. In the case of antioxidant microcapsules, maltodextrin is the most preferred wall material, but others, as Arabic gum, chitosan, pectin, WPI, inulin and zein, are also remarkable. In addition, distinguishing between plant (including most saccharides and vegetable proteins) and animal-based materials (animal proteins, chitosan, gelatin and collagen), a higher percentage of plant-based materials (in more than 65% of the reviewed papers) in comparison to animal-based ones is noted in both fish oil and antioxidant microencapsulation. 

The extensive use of whey protein is related to its excellent surface activity and ability to stabilize oil in water (O/W) emulsions [22]. The most common presentations are WPI, WPC and hydrolysates (WPH). In addition, whey protein coproducts, such as the retentate of the final microfiltration step in the production of WPI (Procream) that mainly comprises dairy lipids and aggregated proteins, have been tested for reutilization and valorization as low-cost emulsifiers in microencapsulation [23]. Maltodextrin has a high solubility in water and can act as a filler matrix to form stable emulsions. Moreover, it has been reported as good protection from oxidation [24]. The use of these polymers is so well established that the current trend is to use them as a control for benchmarking other less-tested materials, alone or in combination, and only a few studies have specifically focused on them. This is the case of some authors [23] who blended Procream with intact or hydrolyzed WPC to improve the microencapsulation efficiency and the oxidative stability of fish oil microcapsules or used WPI conjugated with xylooligosaccharides to encapsulate lycopene, thus enhancing the emulsification performance, the antioxidant capacity, and parameters [25]. Other researchers [26] combined maltodextrin and WPI to microencapsulate fish oil, finding a high oxidative stability of the obtained powders that was attributed to the antioxidant effect of WPI.

Table 1 summarizes the most recent investigations focused on the evaluation of the emulsion constituents for fish oil microencapsulation. Besides the mixtures of whey protein and/or maltodextrin with other wall materials, the evaluation of different polymers of cellulose, inulin, fish protein isolate, chitosan and soy protein isolate has received interest in the most recent studies. García-Moreno et al. [27] showed the potential of different carbohydrates (pullulan and dextran or glucose syrup) and WPC mixtures for the nanoencapsulation of fish oil. In the same way, Charles et al. [28] demonstrated that arrowroot starch, maltodextrin and WPI combinations successfully encapsulated tuna fish oil, and Damerau et al. [29] prepared emulsions of fish oil with WPC and rice proteins as wall materials, obtaining a high stability that was ascribed to the WPC. Jamshidi et al. [30] allowed the stabilization of water in oil in water (W1/O/W2) double emulsions containing fish protein hydrolysate within a complex of WPC and inulin to produce fish oil microencapsulates. Özyurtet al. [31] also tested the use of fish protein isolate in combination with maltodextrin to encapsulate fish oil, finding better quality characteristics in comparison to the sodium caseinate and maltodextrin mixture. Ogrodowska et al. [32] compared different coating materials (maltodextrin, WPC, sunflower and rice proteins and guar gum) for encapsulating fish oil, obtaining the best overall properties when using rice proteins. In addition, the combination of WPC and rice proteins reduced the fishy and rancid odor and flavor of the powder. Chang et al. [33,34] tested the encapsulation of fish oil using different β-lactoglobulin fibril variants from WPI (β-lactoglobulin fibrils and thiol-modified β-lactoglobulin fibrils), chitosan and maltodextrin. These authors demonstrated that combinations of chitosan (a great emulsion stabilizer) and β-lactoglobulin (with high emulsification properties), which are oppositely charged, improve the microencapsulation efficiency. Encina et al. [35] microencapsulated fish oil with hydroxypropylcellulose, due to its solubility in both water and organic solvents, using lecithin as an emulsifier. Hydroxypropylmethylcellulose acetate succinate, which is water-insoluble, was tested by Loughrill er al. [36] to obtain compatible fish oil microcapsules with aqueous-based food products. Encina et al. [37] synthetized hydroxypropyl-inulin by etherification to increase the solubility in water and organic solvents and obtain a novel encapsulating agent for fish oil. Rios-Mera et al. [38] developed fish oil microcapsules with inulin, soy protein isolate and transglutaminase, the enzymatic cross-linking being crucial to improve the retention of fish oil under stress conditions. It is also worth mentioning the use of konjac glucomannan (favorable water solubility and absorption, emulsification and film-forming properties, besides potential health benefits, such as reducing and delaying glucose absorption, inhibiting the synthesis of fatty acids and controlling obesity) [39], alginates (which in addition to enhancing satiety, have the ability to form a gel that preserves the bioaccessibility of the core material, controlling the release of essential fatty acids in specific areas of the gastrointestinal tract) [40], acacia gum (it has emulsifying and film-forming properties and can act as surface-active substance, and once dried, it constitutes a matrix that prevents contact with oxygen) [41], silica (to form a highly organized three-dimensional hybrid matrix nanostructure that enhances bioaccessibility of medium and long chain length triglycerides) [42], carrageenan (a water-holding, gelling, stabilizing, thickening and emulsifying agent) [43] and zein (a prolamine isolated from maize with hydrophobicity, biocompatibility and film-forming properties) [44] as encapsulation materials in recent studies on fish oil microcapsules. Moreover, some recent publications have also been devoted to improving the oxidation stability of fish oil microcapsules by means of including antioxidants together with the wall material or in the core. Thus, Vishnu et al. [45] applied vanillic acid grafted chitosan for the microencapsulation of sardine oil, obtaining promising results. Yeşilsu and Özyurt [46] evaluated the addition of rosemary, thyme and laurel extracts to a previously formed emulsion of anchovy oil with lactose and sodium caseinate, the highest oxidative stability being found in microcapsules with incorporated rosemary and laurel. In the study of Solomando et al. [47], fish oil was mixed with lycopene and microencapsulated by using lecithin and maltodextrin. These authors reported a significant decrease in lipid oxidation markers as the lycopene content increased.

Comparing the obtained results among the different studies might be quite imprecise due to the influence of the encapsulation and analytical procedures. In addition, the quality parameters evaluated are not the same in all reviewed publications. However, in general, the suggested polymers have achieved successful microencapsulation of fish oil with reasonable oxidation stability. 

In most recent studies on natural antioxidant microencapsulation (Table 2), the core material is predominantly constituted by an extract from the leaves, seeds or peels of vegetables (green jelly, red chicory, red cabbage, bay, tomato, sea buckthorn, Securigera securidaca, Japanese quince, Moringa stenopetala, Sida rhombifolia), fruits (olive, cocona, camu-camu, cranberry, pomegranate, araza, mulberry, grape, jabuticaba) or plants (green tea) or even from microorganisms (microalgae). The encapsulation of extracts from rice, oat bran and pepper flour, lycopene, curcumin, propolis, resveratrol, thymol and carvacrol has also been recently addressed. As for the wall material, in several works maltodextrin is blended with different polymers, namely hydrolyzed collagen [48], sodium caseinate and dried glucose syrup [49], Arabic gum [50,51], peel pectin powders [52], cashew gum and Tween 40 [53] and inulin [54]. In others, maltodextrin is benchmarked against oligofructose [14], Arabic gum [55] or inulin [54,55]. In addition, the dextrose equivalent (DE) value of maltodextrin (a measurement of its proportion of reducing sugars) has also been investigated (DE 10-13 vs. DE 17-20) [50]. The different maltodextrin blends reached suitable microencapsulation and protection of antioxidant compounds. Moreover, in the comparison studies, maltodextrin was found to be the best polymer for obtaining high-quality microcapsules. Nevertheless, the use of Arabic gum mixed with maltodextrin (1:1 *w*/*w*) for encapsulating carotenoids with linseed oil as a carrier resulted in considerable degradation during spray drying and during the gastric phase of simulated digestion [51]. The use of zein has also been tested for antioxidant encapsulation. Zein is a corn-derived insoluble protein, only soluble in an aqueous solution of >60% ethanol, capable of encapsulating hydrophobic compounds with low water solubility [56]. Thus, it has been successfully used for the encapsulation of an aqueous extract of green jelly leaf [56], sea buckthorn leaves [57] and thymol [58]. Hydroxypropylmethylcellulose [56], glycosylated WPI [25], alginates [59], soy protein isolate [60], starch [61] gelatin [57] and sucrose [62] have also been individually investigated as encapsulating materials. In addition, different combinations of these wall materials have been explored to encapsulate antioxidants, i.e., alginate, pectin, WPI and sodium caseinate [59]; soy protein isolate with soy soluble polysaccharides and maltodextrin [60]; poly(D,L-lactide-co-glycolide), ethylcellulose and polycaprolactone [26]; gelatin–acacia gum [63]; chitosan–carboxymethylcellulose [63]; chitosan, sodium alginate and Arabic gum [64]; carrageenan, lupin protein isolate and chitosan [65]; and WPI and acacia gum [66]. In general, the microencapsulation and protection of the antioxidants have been successfully achieved in these investigations.

Thus, for both fish oil and antioxidant microencapsulation, the current trend seems to be the comparison and/or combination of more-tested wall materials, mainly maltodextrin and whey proteins, with less-tested ones, most of them being from plants.

**Table 1 foods-11-03291-t001:** Evaluated effect, related to the emulsion constituents, the procedure and the food enrichment, in recent studies for the development of fish oil microcapsules.

Effects	Microencapsulation Technique	Reference
Constituents of Emulsion for Microencapsulation		
Use of Procream with intact or hydrolyzed whey protein concentrate	Spray drying	[23]
Use of octenyl succinic anhydride modified starch, gelatin or whey protein isolate with maltodextrin	Spray drying	[26]
Use of whey protein concentrate in combination with pullulan and dextran or glucose syrup	Electrospraying	[27]
Use of arrowroot starch, maltodextrin and whey protein in different combinations	Freeze drying	[28]
Use of rice and whey protein concentrate	Spray drying	[29]
Use of fish protein hydrolysate, whey protein concentrate and inulin	Spray drying	[30]
Use of maltodextrin, thiol-modified β-lactoglobulin fibrils and chitosan	Spray drying	[33,34]
Use of maltodextrin, whey proteins, sunflower proteins, rice proteins, guar gum	Spray drying	[32]
Use of hydroxylpropylmethylcellulose acetate succinate	Spray drying	[36]
Use of hydroxypropylcelullose and hydroxypropyl-inulin	Spray drying	[37]
Use of soy protein isolate, inulin and cross-linked transglutaminase	Complex coacervation + sieving	[38]
Use of konjac glucomannan, soybean protein isolate and potato starch	Spray drying/freeze drying	[39]
Use of alginate calcium	Extrusion with calcium chloride	[40]
Use of skimmed milk powder, acacia gum, and a mixture of acacia gum and grape juice	Spray drying + spray chilling	[41]
Use of silica	Spray drying	[42]
Use of myofibrillar protein in the presence, or not, of κ- or λ-carrageenan	Spray drying	[43]
Use of zein	Electrospraying assisted by pressurized gas	[44]
Use of vanillic acid grafted chitosan	Spray drying	[45]
Comparison between fish protein isolate–maltodextrin and sodium caseinate–maltodextrin	Spray drying	[46]
Use of natural plant extracts (thyme, rosemary, and laurel) in comparison to synthetic antioxidant (BHT)	Spray drying	[46]
Use of lycopene as antioxidant	Spray drying	[47]
Comparison between fish protein isolate–maltodextrin and sodium caseinate–maltodextrin	Spray drying	[46]
Procedure		
Percentage of whey protein concentrate, hydrolyzed whey protein concentrate and Procream	Spray drying	[23]
Emulsification by means of high-pressure homogenization or rotor–stator	Electrospraying	[27]
Optimization of emulsion pH and number of homogenization steps	Spray drying	[29]
Optimization of high-pressure homogenization conditions (pressure and number of cycles)	Spray drying	[30]
Optimization of fish oil/hydroxypropylcellulose ratio and the air inlet temperature	Spray drying	[35]
Direct spray drying vs. production of solid lipid nanoparticle dispersions + spray drying	Spray drying	[36]
Optimization of fish oil:hydroxypropyl-inulin ratio and the inlet gas temperature	Spray drying	[37]
Spray drying vs. freeze drying	Spray drying/freeze drying	[39]
Optimization of oil load	Extrusion with calcium chloride	[40]
Combination of spray drying and spray chilling	Spray drying + spray chilling	[41]
Optimization of pH and ratio between myofibrillar protein and κ-carrageenan or λ-carrageenan	Spray drying	[43]
Optimization of high-pressure homogenization conditions (pressure and number of cycles)	Spray drying	[67]
Impact of oil load	Spray drying	[68]
Optimization of soy protein isolate:oil ratio; emulsification by ultra-turrax vs. ultrasonics	Spray drying	[69]
Vacuum spray-drying technology in comparison to spray drying	Vacuum spray-drying	[70]
Optimization of temperature, pressure and feed flow rate	Supercritical antisolvent process	[71]
Food Enrichment		
Incorporation into melted chocolate that covers extruded cereals	Spray drying + spray chilling	[41]
Incorporation into reconstituted milk	Electrospraying assisted by pressurized gas	[44]
Incorporation into spreads	Spray drying	[47]
Incorporation into burgers	Freeze drying	[72]
Incorporation into cooked and dry-cured sausages	Spray drying	[72,73,74,75,76,77]

**Table 2 foods-11-03291-t002:** Evaluated effect, related to the solution constituents, the procedure and the food enrichment, in recent studies for the development of natural antioxidant microcapsules.

Effects	Core Material	Microencapsulation Technique	Reference
Constituents of Emulsion/Solution for Microencapsulation			
Use of maltodextrin, inuline and oligofructose	Camu-camu extracts	Spray drying	[14]
Use of whey protein isolate glycosylated with xylo-oligosaccharides	Lycopene	Freeze drying	[25]
Use of poly(D,L-lactide-co-glycolide), ethyl cellulose and polycaprolactone	Propolis	Solvent evaporation technique	[26]
Use of maltodextrin and hydrolyzed collagen	Cocona pulp	Spray drying	[48]
Use of sodium caseinate, maltodextrin and dried glucose syrup	Resveratrol	Spray drying	[49]
Use of Arabic gum and maltodextrin (DE 10-13 and 17-20)	Cranberry extract	Spray drying	[50]
Use of maltodextrin and Arabic gum	Tomato peel extract	Spray drying	[51]
Use of cashew gum, maltodextrin and Tween 40	Green tea extract	Spray drying	[53]
Use of maltrodextrine and/or inulin	Japanese quince juice	Spray drying/freeze drying/vacuum drying	[54]
Use of maltodextrin or Arabic gum	Araza pulp	Spray drying	[55]
Use of zein and hydroxypropylmethyl cellulose	Green jelly leaf extract	Spray drying	[56]
Use of maltodextrin and peel pectin powders	Pomegranate peel extract	Spray drying	[56]
Use of zein or gelatin	Sea buckthorn leaf extract	Electrohydrodynamic method	[57]
Use of zein	Thymol	Spray drying	[58]
Use of alginate alone or in combination with pectin, whey protein isolate and sodium caseinate	Olive leaf extract	Internal gelation	[59]
Use of soy protein isolate nanocomplexes combined, or not, with soy soluble polysaccharide and/or maltodextrin	Curcumin	Spray drying	[60]
Use of modified starch (CAPSUL)	Red chicory and red cabbage extract	Spray drying	[61]
Use of sucrose	*Securigera securidaca* seed extract	Dried in oven	[62]
Use of gelatin–acacia gum or chitosan–carboxymethyl cellulose	Black rice extract in combination or not with copigmented anthocyanins	Freeze drying	[63]
Use of modified chitosan, sodium alginate and Arabic gum	Bay leaf extract	Spray drying	[64]
Use of carrageenan, lupin protein isolate and chitosan	Astaxanthin oleoresin	Spray drying	[65]
Use of whey protein isolate and acacia gum	Tomato peel extract	Freeze drying	[66]
Procedure			
Different maltodextrin:Arabic gum ratio	Red cabbage anthocyanin-rich extract	Double drum dryer	[21]
Different propolis extract:polymer ratios	Propolis	Solvent evaporation technique	[26]
Optimization of membrane emulsification process conditions	Resveratrol	Spray drying	[49]
Optimization of proportion of wall material (maltodextrin or Arabic gum) and drying temperature	*Eugenia stipitata* pulp	Spray drying	[55]
Spray drying vs. freeze drying vs. vacuum drying	Japanese quince	Spray drying/freeze drying/vacuum drying	[57]
Optimization of inlet air temperature, type of emulsion, feeding rate and total solids	Astaxanthin oleoresin	Spray drying	[65]
Different combinations of total solid content, emulsifier, carrier agent and drying method	Fucoxanthin-rich fraction from microalgae	Spray drying/freeze drying	[71]
Impact of maltodextine:high methoxyl pectin ratio. Spray drying vs. freeze-dying	*Moringa stenopetala* leaf extract	Spray drying/freeze drying	[78]
Different whey protein concentrate:maltodextrin ratios	Oat bran extract	Complex coacervation	[79]
Optimization of maltodextrin:Arabic gum ratio, maltodextrin dextrose equivalent, relative humidity and time during storage	Grape pomace extract	Spray drying	[80]
Optimization of percentage of solids, maltodextrin:trehalose dehydrate ratio and olive leaf extract:matrix ratio	Olive leaf extract	Freeze drying	[81]
Maltodextrin concentration	Pomegranate flavedo extract	Lyophilization	[82]
Amount of sodium alginate and volume of the extract	Araza extracts	Drip extrusion	[83]
Different homogenization techniques. Optimization of core material:wall material ratio, ultrasonic time, ultrasonic power and ultrasonic temperature	Mulberry polyphenols	Freeze drying	[84]
Impact of extract concentration	*Sida rhombifolia* extracts	Spray drying	[85]
Optimization of dispersion feed rate, drying air inlet temperature and drying air flow rate	Grape peel by-product extract	Spray drying	[86]
Effect of inlet air temperature	Carvacrol	Spray drying	[87]
Effect of inlet air temperature	Pepper flour extract	Spray drying	[88]
Food Enrichment			
Incorporation into dressing	Tomato peel extract	Freeze drying	[66]
Incorporation into a fruit drink	Pomegranate flavedo extract	Lyophilization	[82]
Incorporation into cassava starch biscuits	Jabuticaba extracts	Drip extrusion	[83]

## 4. Procedure for Emulsion/Solution Preparation

In general, the procedure for obtaining a powder of encapsulates involves the preparation of an emulsion or solution followed by its desiccation. The aim of the preparation of the emulsions or solutions is obtaining a uniformly consistent mixture between core and wall materials. This is a key step, especially in the case of emulsions. Magnetic stirring and/or rotor–stator systems are normally used for preparing fish oil emulsions. In addition, in some works, an additional homogenization step is also applied aiming for improving emulsion stability and, consequently, microencapsulation efficiency and the overall quality of the microcapsules. In the latest studies, this complementary procedure has been increasingly applied, in most cases by means of high-pressure homogenizers [36,68,69]. Recently, a microfluidizer and a vacuum emulsification machine were also used for high-pressure homogenization of fish oil emulsions [83,84]. The high-pressure homogenization step aims to reduce and standardize the size of oil particles, by breaking aggregates of particles and spreading them evenly. However, it may also have undesired side effects, such as increase in viscosity, foam formation, changes in the surface area, and even increased lipid oxidation. Therefore, an optimization of the processing parameters, mainly pressure and number of cycles, is necessary when using high-pressure homogenizers [67]. The use of ultrasounds has also been applied for the extra homogenization stage of fish oil emulsions in some recent studies [69,70]. This technique favors the dispersion of the oily phase, increases the solubility of wall materials, and reduces the viscosity. In the case of antioxidants, usually, a solution including an aqueous extract and the wall materials is prepared by magnetic stirrer and/or rotor–stator systems. 

The optimization of different parameters of the emulsion/solution preparation has been addressed in several of the recent works on fish oil (Table 1) and antioxidant microencapsulation (Table 2). As can be seen in Figure 3, the wall material ratio in the solutions has been extensively investigated. Thus, the total solid content effect on the microencapsulation of the fucoxanthin-rich fraction extracted from microalgae was evaluated, the highest antioxidant activities being found with 40% of wall material (1:1 maltodextrin:Arabic gum) and 1% Tween 20 [71]. In the work of Dadi et al. [78], the maltodextrin:high methoxyl pectin ratio (10:0 vs. 9:1 (*w*/*w*) significantly influenced most quality characteristics of microcapsules of Moringa stenopetala leaf extract, the 9:1 ratio being the most efficient one. In the microencapsulation of oat bran extract with different WPC:maltodextrin ratios [79], the 60:40 ratio showed the highest microencapsulation efficiency. Similarly, Tolun et al. [80] and Sakulnarmrat et al. [21] optimized the maltodextrin:Arabic gum ratio for the microencapsulation of grape pomace red cabbage extracts, with the two studies finding opposite ratios as optimum results (8:2 and 2:8, respectively). In the study of González-Ortega et al. [81], the microencapsulation efficiency of olive leaf extract was found to be positively affected by higher concentrations of maltodextrin, while the percentage of total solids did not show any influence. Iturri et al. [55] also observed a positive influence of maltodextrin concentration in the preservation of polyphenols in Eugenia stipitata pulp microcapsules. On the contrary, Hamid et al. [82] found better functional properties and structural morphology in microcapsules of pomegranate flavedo extracts with lower concentrations of maltodextrin, and Morales et al. [65] found a scarce effect of maltodextrin concentration on the microencapsulation efficiency of astaxanthin microcapsules. In the study of de Cássia Sousa Mendes et al. [83], the concentration of sodium alginate did not affect the microencapsulation efficiency of jabuticaba extract microcapsules. In recent studies on fish oil microcapsules, Bakry et al. [43] concluded that 4:1 was the optimal myofibrillar protein:carrageenan ratio for encapsulation, and Hinnenkamp et al. [23] found a positive effect of whey protein content on the quality of fish oil microcapsules, with the highest microencapsulation efficiency and oxidation stability found at 2%.

The influence of the device and conditions for the emulsion/solution preparation and the core and core:wall material ratios and pH have also been a matter of interest, especially in studies dealing with fish oil microcapsules. A positive effect of applying rotor–stator plus high-pressure or ultrasonic homogenization has been evidenced in fish oil microcapsules [27,69]. In addition, other authors have optimized high-pressure homogenization conditions (pressure and number of cycles) in simple, multilayered and doubled emulsions of fish oil [67,71,89]. In microcapsules of mulberry extract, Li et al. [84] evaluated the use of rotor–stator, magnetic stirring, grinding, ultra-high-pressure and ultrasonic methods; these authors selected ultrasound homogenization as the preferred method and further optimized the time, power and temperature sonication conditions. Consoli et al. [49] optimized an innovative emulsification technique for developing resveratrol microcapsules. It involved using a dispersion cell with the continuous phase being placed within a glass cylinder, into which the dispersed phase is pumped through a flat metal membrane placed at the bottom. Stirrer speed, shear stress and feed rate were the optimized variables, and the optimization achieved notably improved microcapsule quality parameters, especially microencapsulation efficiency, mean diameter and water content. Regarding the core ratio, Bannikova et al. [40] established a maximum of 15% oil load to obtain a controlled release under simulated gastrointestinal conditions, and in the study by Linke et al. [68], an increase in the amount of oxidation products due to higher oil load was evidenced, especially from 10% and upwards. However, in Sida rhombifolia extracts, an increase in oil load from 2.5 to 5% did not lead to important differences in the quality of the microcapsules [85]. The ratio between core and wall material (hydroxypropylcellulose, hydroxypropyl-inulin and soy protein isolate) has also been optimized in recent studies on fish oil [35,37,75] and antioxidant microcapsules [76,77]. Most of these works point to a positive effect of a higher core:wall material ratio on the microencapsulation efficiency. The effect of pH in fish oil emulsions prepared with a blend of rice and whey protein was evaluated by Damerau et al. [29], and microcapsules with better quality characteristics were obtained with emulsions at pH 6.5, while in the study by Bakry et al. [43], the optimum fish oil microencapsulated with myofibrillar proteins and carrageenan was achieved at pH 5. It seems that the optimum pH depends on the wall material due to its strong influence on the degree of ionization of the functional groups (carboxyl and amino groups) as well as the strength of electrostatic interactions between the macromolecules that carry electrical charges [13].

## 5. Procedure for Microencapsulation

In relation to the microencapsulation procedure, Figure 4 shows some of the different techniques described in recent studies for the production of fish oil and antioxidant microcapsules. Spray drying was the preferred technique in these studies, but the use of freeze drying is also remarkable. Spray drying is a physicomechanical method commonly used for microencapsulation. It is simple, flexible, rapid and easy to scale up; has low operating cost; and allows a large-scale production in continuous mode. Moreover, spray drying has a wide range of equipment availability, and it is considered a clean technology (avoids the utilization of organic solvents). It consists in transforming a liquid solution into powders by means of different steps: atomization of the previously prepared liquid solution (by dissolving or emulsifying the active compounds (core) with the carriers (wall material)) leading to the formation of droplets a few microns in diameter; spraying of the droplets in a heat chamber where they are dehydrated into capsules by applying hot air (160–200 °C) for a few seconds; separation of the capsules from the drying air at a lower temperature (50–80 °C) and recovery of the capsules [90]. It is a continuous and single process where the moisture is rapidly evaporated, and this allows keeping a relatively low temperature in the particles. The obtained powder has a very low water activity, ensuring microbiological stability, and it is easy to handle, store and transport. Spray-chilling and spray-cooling techniques work in a similar way to spray drying, but instead of evaporation of the water of the emulsion/solution, a hardening of the surface wall material is achieved. In these processes, the solution is atomized and droplets are cooled instead of evaporated. Thus, the wall material solidifies around the core. Like spray drying, spray chilling and spray cooling are rapid, safe, reproducible, easy to scale up and environmentally friendly and can be operated continuously. However, they also have high process costs and special handling and storage conditions since the obtained material is not as shelf-stable as those produced by spray drying [90]. Freeze drying or lyophilization is a drying method that dehydrates the solution of core and wall material mixtures, previously frozen, by sublimation under vacuum and low temperatures. The obtained products present a well-reduced risk of the undesirable effects derived from heating and are easily reconstituted. However, freeze drying is very expensive and time-consuming due to the longer dehydration process, the vacuum technology applied and the low temperatures required [4].

Most of the recent studies that focused on the microencapsulation procedure evaluated the influence of different parameters and compared techniques (Figure 3). For example, Carra et al. [86] prepared microcapsules of grape extracts by spray drying and investigated the role of dispersion feed rate, drying air inlet temperature and drying air flow rate on several quality parameters, the inlet temperature being the most influential factor. In the study by Sun et al. [87], four inlet temperatures (100, 130, 160 and 190 °C) were tested for the encapsulation of carvacrol; these authors found that inlet temperatures between 100 and 130 °C were the most appropriate in terms of antioxidant capacity, dissolution time, hygroscopicity and morphology. de Sá Mendes et al. [88] also studied the consequences of inlet temperature (140 to 160 °C) on the physicochemical properties of spray-dried pepper flour extract, concluding that color and morphology were strongly influenced. Brighter, less purple and spherical particles with some shrinkage were obtained with high inlet temperatures. Cui et al. [39] evaluated the properties of fish oil microcapsules prepared by either spray drying or freeze drying. In general, these authors found better overall quality in microcapsules prepared with spray drying (higher microencapsulation efficiency, more spherical particles with a compact structure, higher retention rate of core materials). Dadi et al. [78] compared the use of spray drying and freeze drying for microencapsulating an extract from Moringa stenopetale leaves. Although the phenolic and flavonoid contents and antioxidant activity were higher in freeze-dried microcapsules, these authors chose the spray-drying technique since the microencapsulation efficiency and storage stability were higher. Similarly, Turkiewicz et al. [54] tested freeze drying, spray drying and vacuum drying for the encapsulation of Japanese quince juice. For most of the analyzed parameters in the powders, freeze drying had a positive effect, while vacuum caused deterioration of all parameters. The combination of different microencapsulation technologies has also been tested. Thus, Fadini et al. [41] suggested the application of spray-drying and spray-chilling techniques for producing fish oil microcapsules with no fish oil taste. A first shell to cover the fish oil is formed by spray drying; then, spray chilling is applied to create a second shell, using the spray-dried microcapsules as the core and a mixture of vegetable fat and hydrogenated palm oil as the wall material. Solvent and vacuum spray-drying techniques were recently evaluated for the production of fish oil microcapsules in order to avoid the increase in lipid oxidation during spray drying. Solvent spray-drying avoids the preparation of a fish oil-in-water emulsion, and vacuum spray-drying can overcome the high temperature and high oxygen exposure due to the airflow, both of which boost oxidative reactions. Encina et al. [35] studied the use of solvent spray-drying with ethanol, methanol and acetone. Although results have shown that solvent spray-drying affects microencapsulation efficiency and stability in different ways, this technique has the potential to be an alternative technology for the encapsulation of fish oil. In the case of vacuum spray-drying, it significantly improved the quality properties of fish oil microcapsules in comparison with spray-dried ones [91]. Nevertheless, these authors indicated the need of further studies with these techniques.

Complex coacervation, electrospraying, supercritical antisolvent, extrusion and internal gelation have also been recently applied as encapsulation techniques by some authors. The complex coacervation technique is based on homogenizing two aqueous solutions, one with the bioactive compound mixed with an anionic polymer and the other containing a cationic polymer, to obtain a stable emulsion, which is induced to coacervate and undergo a phase separation by pH and/or temperature modifications. The coacervates are formed by larger particles of higher density at the bottom of the two-phase system. The major limitation of this technique is the high sensitivity to pH and ionic strength of the coacervates, which makes its commercial application difficult [13] Rios-Mera et al. [38] formed complex coacervates by adding an inulin solution to a previously developed fish oil–soy protein isolate emulsion. In this study, the obtained microparticles were sieved and cross-linked with transglutaminase. Bannikova et al. [79] added a guar gum solution to a mixture of oat bran extract with maltodextrin and WPC and dried it to produce the microcapsule powder.

Extrusion techniques involve forcing a liquid mixture (core and wall materials) through an orifice to form droplets at the discharge point of the nozzle. After the droplets are formed, capsules are immediately solidified by physical or chemical processes. In coextrusion, core and wall materials are injected separately in a double fluid nozzle [8]. Apart from coextrusion, there are other different types of extrusion techniques, melt injection, hot-melt extrusion and electrostatic. There are some recent works that achieved the microencapsulation of fish oil and natural antioxidants by means of the extrusion technique. Bannikova et al. [40] applied this technique to microencapsulate fish oil by preparing an emulsion with alginate that was dripped into calcium chloride followed by sieving, with the obtained microcapsules being stable during the gastric stage and released at the intestinal phase. Similarly, de Cássia Sousa Mendes et al. [83] dripped a mixture of sodium alginate with an extract of jaboticaba peel and seeds in a calcium chloride solution and obtained microcapsules with high encapsulation efficiency and antioxidant potential. 

Two main electrodynamic techniques are used to encapsulate bioactive compounds, electrospraying and electrospinning. Both involve the application of an electrical field to charge the polymer solution, which is subsequently induced to free charges and forms nanofibers (electrospinning) or is atomized into droplets, giving charged nanodroplets (electrospraying). Despite being novel techniques, both processes are slow, difficult to scale up and have low yield and efficiency, which limit their commercial exploitation for foods [4]. Busolo et al. [44] and García-Moreno et al. [27] applied electrospraying to microencapsulate fish oil by using whey protein carbohydrates and zein as wall materials, respectively, with stable microcapsules being attained; Lyu et al. [57] tested this technology for encapsulating polyphenols and flavonoids from the leaves of sea buckthorn, resulting in successful protection from degradation and bioaccessibility in the intestine.

The use of supercritical fluid processes to encapsulate bioactive compounds is gaining attention. This technique is based on the peculiar properties of supercritical fluid, which is an intermediate between liquid and gas and can be easily changed with modifications in its pressure and temperature. Depending on the role that the supercritical fluid plays in microcapsule formation, this technique can be classified as solvent (the target compound and the wall material are dissolved in a supercritical fluid at high pressure; once the supercritical fluid is expanded, pressure is reduced and precipitation of the particles occurs), antisolvent (the target compound and the wall material are dissolved in a liquid solvent and atomized together with the supercritical fluid, which decreases the solubility of the solute in the mixture, with the consequent formation of nano- or microparticles) or solute (the supercritical fluid is dissolved into a suspension of the target compound and wall material, followed by a rapid depressurization of the solution through a nozzle and the final formation of solid particles or liquid droplets) [4]. By submitting the fluids to high pressure, this technique allows applying low temperatures, small solvent volume and sample quantity, and short extraction times. In addition, this method eliminates or reduces the solvents due to the high solubility of the solvent in the supercritical fluid, which is easily removed from the final product by depressurization. However, due to the high pressure variations applied, this technique implies a high cost. Based on their findings on particle size, moisture content, surfaces, microencapsulation efficiency, stability against oxidation and other microcapsule quality parameters, Karim et al. [92] pointed out the capability of the supercritical antisolvent process in encapsulating fish oil for industrial application using hydroxypropyl methyl cellulose as a wall material.

The ionotropic gelation method is based on the interaction between ionic polymers and ions with opposite charges that can be positioned externally or incorporated within the polymer solution in an inactive form, giving rise to external and internal ionotropic gelation methods, respectively. In both cases, hydrogel microbeads are produced. Flamminii et al. [59] achieved the microencapsulation of olive leaf extract through internal gelation by using alginate alone or mixed with pectin, whey proteins or sodium caseinate. The combination of alginate with pectin achieved the highest microencapsulation efficiency and retention of phenolic compounds.

Moreover, in some recent studies for antioxidant microencapsulation, simple solvent evaporation has been used, as reported by Paulo et al. [26], who microencapsulated propolis extract after evaporating the solvents of a double emulsion in a fume hood at room temperature. Behnamnik et al. [62] encapsulated Securigera securidaca seed extract using sucrose and heating to 132 °C to co-crystalize the product, which was subsequently dried in an oven. In the work of Sakulnarmrat et al. [21], cabbage anthocyanin-rich extract was encapsulated by mixtures of maltodextrin and Arabic gum using a double drum dryer. In these studies, the obtained microcapsules showed high antioxidant properties.

## 6. Microcapsule Quality Evaluation

Figure 5 shows the extent of use in recent publications of most applied analytical determinations for the quality evaluation of fish oil (Figure 5A) and antioxidant microcapsules (Figure 5B). More than 40% of studies evaluated particle size and morphology, microencapsulation efficiency and moisture of the final powder, as well as oxidation stability in the case of fish oil microcapsules and antioxidant activity, solubility/solution/hygroscopicity and release/stability for antioxidant microcapsules. A lower percentage (<35%) of studies analyzed water activity, Fourier transform infrared spectroscopy (FTIR), instrumental color, in vitro digestion, differential scanning calorimetry (DSC), flowing properties, yield and pH. Some studies on fish oil microcapsules also evaluated the fatty acid composition (around 35%) and hygroscopicity, solubility, release and reconstitution properties (<15%).

In most studies, microcapsule size was determined by laser diffraction using laser light diffraction instruments, in a dry dispersion [67] or by dissolving the samples in water [58,59] or solvents such as ethanol [77,91,93] and acetic acid [39]. Results of this type of analysis are usually shown by means of particle size distribution curves (percentage volume vs. diameter) and mean diameter. In these recent studies, most evaluated variables evidenced an influence on these quality parameters, having a fairly broad range in the size of microcapsules, i.e., fish oil spray-dried microcapsules showed a mean diameter of 29.90 μm, which was smaller than that of freeze-dried microcapsules (193.55 μm) (Cui et al., 2021). Solomando et al. [67] observed the influence of the homogenization of fish oil emulsions with high pressure on the size homogeneity of the microcapsules, with high-pressure microcapsules presenting a monomodal distribution, while those without high-pressure homogenization showed a bimodal distribution with a wider distribution than homogenized ones. Similarly, the use of different wall materials also led to differences in the particle size distribution, with gelatin + maltodextrin showing a wider range than octenyl succinic anhydride modified starch + maltodextrin and whey protein isolate + maltodextrin [26]. In the study of Flamminii et al. [59] with microcapsules of olive leaf phenolic extract, the use of alginate–whey protein isolate led to microcapsules with smaller size in comparison to those resulting from the use of alginate alone or mixed with pectin or sodium caseinate. An automatic standard sieve shaker has also been used to calculate the size of microcapsules [78], and Ferro et al. [85] have calculated the size of microcapsules from scanning electron microscopy images using the ImageJ software. The microcapsule size may determine the pertinence for application. A mean diameter of around 10 μm seems to be convenient for food addition without influencing mouthfeel attributes, with the acceptable limit being around 30 μm [49]. The bimodal distribution is interesting for packaging because the smaller microcapsules can be between the larger ones, reducing the occupied space, the included air within the particles and thus the probability of lipid oxidation [91].

Table 3 details the most used techniques for the quality evaluation of fish oil and antioxidant microcapsules. In most studies, the microencapsulation efficiency has been determined as a function of the quantity of encapsulated bioactive compound to the total content of the microcapsules, with the external and total contents being determined (microencapsulation efficiency = (total quantity − external quantity)/total quantity x 100). Nevertheless, there are some studies considering the total quantity as the initial quantity added to the emulsion or dissolution [38,59,70]. In the studies of fish oil microcapsules, external oil is normally extracted with an organic solvent such as petroleum ether or hexane, while for the total oil extraction, a hydrolysis step, normally with chloridric acid, is needed before the oil recovery. In the work of Linke et al. [68], a nuclear magnetic resonance (NMR) analyzer was used to measure the amount of total and encapsulated oil. In the case of the antioxidant microcapsules, the specific compounds (lycopene, polyphenols, anthocyanins, etc.) are quantified. The major aim in the microencapsulation of bioactive compounds is to increase the microencapsulation efficiency as much as possible; however, it is a highly variable parameter, as observed in the results shown in most recent works in fish oil (i.e., 71.1 and 92.0% in fish oil microcapsules prepared with conventional spray-drying and solvent spray-drying with acetone, respectively [35]; 88.88 and 46.72% in fish oil microcapsules from emulsion with and without high-pressure homogenization, respectively [67]) and antioxidant microcapsules (i.e., 71.44 and 83.52% in freeze-dried and spray-dried microcapsules of Moringa stenopetala leaf extract [78]; 95.3 and 77.5% in oat bran extract microcapsules with 60/40 and 70/30 whey protein–maltodextrin ratio, respectively [79]).

Scanning electron microscopy has been used to observe the morphology of the microcapsules of fish oil and antioxidants. Spherical defined particles with smoother surfaces with a good degree of integrity are signs of stability of the microcapsules. The absence of pores in the skin of the microcapsules has been highlighted as an advantage, since it ensures better protection and retention of the encapsulated compounds, as the presence of pores can lead to an increase in the permeability of the wall material, decreasing the protective effect of the core [94]. The presence of holes in the surface of microcapsules indicates instability and confirms the lack of integrity. Additionally, the holes allow the entry of oxygen into the capsules and the exit of core materials, thus increasing oxidation and decreasing product quality. Collapse and shriveling are not very desired and have been related to the composition of encapsulating material and drying parameters. Skin-forming particles, crystalline particles and agglomerate particles are the three distinct morphologies in which spray-dried microcapsules can be found [95]. In spray-dried microcapsules, visible wrinkles or dimples on the surface are sometimes detected, being attributed to rapid evaporation of the drops during atomization in the drying process. In fact, Dadi et al. [78] have observed differences in the morphology between spray-dried (spherical shape with dented surface) and freeze-dried (glassy and irregular shapes with edges) microcapsules of Moringa stenopetala leaf extract. Solomando et al. [67] have found the influence of the high-pressure homogenization of fish oil emulsion on microcapsule morphology: homogenized microcapsules showed an egg-shape morphology with a smooth surface, with shriveling but no pores, wrinkles or dimples, while non-homogenized ones were spherical and presented pores and dimples and even some broken microcapsules. In the study of Hamed et al. [70] with fish oil microcapsules, the addition of curcumin as a natural antioxidant led to a less smooth surface with visible wrinkles and agglomerates as compared to microcapsules with the addition of curcumin.

Moisture has been usually analyzed by following the AOAC method (reference 935.29) [96], by drying the powders in an oven at 105 °C for at least 2 h and calculating the percentage of moisture gravimetrically. However, in some recent studies, moisture analyzers with continuous heating until achieving a constant mass value [28,92] or infrared [35] have also been used. Determination of moisture in microcapsules is required because a low moisture percentage (<4%) is related to stability against microbiological spoilage and preservation of physicochemical properties, which guarantees long shelf life during storage and final use [89]. 

In the studies on fish oil microcapsules, oxidation stability is a chief quality parameter. Usually, primary and/or secondary oxidation products are determined, mainly by means of peroxide value and thiobarbituric active reactive substance (TBARS) methods, respectively. The profile of volatile compounds has been also analyzed in some works by headspace solid phase microextraction (HS-SPME) and gas chromatography with mass spectrometry (GC-MS) detection [29,83]; of special interest are volatile compounds from omega-3 fatty acid oxidation with rancid and off-flavor perceptions, such as 2,4-heptadienal, 2,4-decadienal, 2-nonenal, 3,5-octadien-2-one and 1-octen-3-one, as investigated by Solomando et al. [73]. In the fish oil microcapsules evaluated by these authors (with maltodextrin and maltodextrin–chitosan), these volatile compounds were not detected, indicating the effectiveness of the wall materials used to minimize the contact and reactivity of encapsulated fish oil with oxidizing promoters. Moreover, attenuated total reflection Fourier transform infrared (ATR-FTIR) [27] and proton nuclear magnetic resonance (1H NMR) [42] techniques have also been applied to determine the oxidation stability in fish oil microcapsules. The use of ATR-FTIR was based on the intensity of the characteristic absorption band of omega-3 PUFAs at 3012 cm-1, which corresponds to the stretching of cis-alkene (-HC=CH-) groups, indicating their disappearance or not. The decrease in the frequency of the band at 3006–3012 cm^−1^ is associated with advanced stages of lipid oxidation. In fish oil electrosprayed microcapsules, the absorbance of the cis-alkene groups did not decrease, which implies their oxidation stability [27]. 1H NMR focuses on comparing the relative intensities of key peaks associated with the polyunsaturated alkyl chain. An intensity reduction and/or absence of key peaks related to oxidized fish oil indicates oxidation stability, as found by Joyce et al. [42] for porous silica microcapsules of fish oil.

Most recent studies on microencapsulation of natural antioxidants have applied the 2,2-diphenyl-1-picrylhydrazyl (DPPH) free-radical scavenging method to evaluate the antioxidant activity, measuring the capacity of the microcapsule to inhibit free radicals based on color changes that can be determined spectrophotometrically. It is normally calculated by percentage, relating the absorbance of samples and blank, although it has also been quantified from a standard curve (generated with Trolox or ascorbic acid) and expressed as mg Trolox or ascorbic acid equivalent per gram of microencapsulate [7,57]. Besides this method, some studies have also carried out the ferric-reducing antioxidant-power (FRAP) [7,71], the ferrous-ion chelating activity [78], the hydroxyl radical scavenging activity [25], the 2,2′-azino-bis(3-ethylbenzthiazoline-6-sulfonic acid) (ABTS) scavenging [71,83], the metal chelating activity [82] and the oxygen radical absorbance capacity (ORAC) [60] assays.

Determining the solubility of microencapsulates is essential when incorporation into foods is considered, and solubility is also a key parameter for improving the bioaccessibility of the encapsulated bioactive compounds. It is usually determined by dissolving the powder in water and subsequently centrifuging and drying the supernatant and calculating the percentage of solubility gravimetrically [7,21,25,65]. With the same objective, other authors have determined the time it takes for the powder in contact with water to completely dissolve [87]. In addition, some works have evaluated the hygroscopicity of the powders, defined as the capacity of a product to absorb moisture from the surrounding environment. It is normally measured as the moisture content (grams of water absorbed by 100 g of the powder) after the product has been placed for a week in a desiccator with saturated sodium chloride solution (at a relative humidity of 75%) [7,21,87]. Relatively low hygroscopicity percentages (<10%) are desirable. Results of different studies have demonstrated that these parameters are greatly influenced by several variables, such as the methodology and parameters for desiccation or the wall material [7,21,25,87].

To evaluate the release or stability of the encapsulated antioxidant compounds, in recent works, the powders have been subjected to different conditions; i.e., the authors of [25] determined the retention rate of lycopene encapsulated in WPI–xylo-oligosaccharide conjugates by storing the samples for 36 days at 4°C, 25°C and 40°C in the dark; in thymol-loaded zein microcapsules, the authors of [58] performed a release test with dialysis bags at pH = 7.4 and mild magnetic stirring; the release of antioxidants from zein microcapsules of green jelly leaf extract has been assessed at different pH levels (3 and 7 to simulate pH conditions in juices and water) in darkness at room temperature [56]; the stability of cocona pulp microencapsulated with a hydrolyzed collagen and maltodextrin blend was studied under controlled (25 ± 2 °C) and accelerated (35 ± 2°C) conditions for 120 days in an incubator to ensure minimum exposure to light [48]; Behnamnik et al. [62] applied different environmental conditions (25 °C in the dark and in artificial light, and refrigeration at 4 °C) for four months to evaluate the stability of powders of Securigera securidaca seed extracts. 

Some recent works on fish oil and antioxidant microcapsules have also applied FTIR, principally to evaluate the interactions between the wall material and core. For example, in the study of Li et al. [84], characteristic bands for mulberry polyphenols and cyclodextrin analyzed separately and mixed were found to be different from the peak profile of the corresponding microcapsules. Hamed et al. [70] encapsulated fish oil with alginate–chitosan and curcumin as a natural antioxidant, obtaining similar FTIR spectra in microcapsules with and without curcumin, which indicated the absence of chemical interactions between curcumin and the components of the microcapsules. 

The analysis of the microcapsules by means of differential scanning calorimetry (DSC) serves to detect processes such as melting, solid–solid transition, dehydration and glass transitions, providing information about the physical behavior of materials. Glass transition temperature is normally measured; it is defined as the temperature in which amorphous products change from a glassy structure (solid) to a rubbery structure (liquid-like). Generally, a glassy shell is desired in order to prevent the diffusion of oxygen to the inner section of the microcapsules. In fish oil microcapsules of thiol-modified β-lactoglobulin fibril–chitosan complex, glass transition temperature increased with the increase in inlet temperature, which has been related to the instantaneous formation of a solid dry surface at a high inlet temperature that may restrain the liquid bridges between the particles and, consequently, suppress the plasticizing activity of water [34]. García-Moreno et al. [27] found differences in glass transition between fish oil microcapsules made of glucose (glass transition at 94.2) and dextran, which did not show glass transition temperature, implying that the wall material will be in a glassy state up to 200 °C. 

As for the flowing properties of microcapsules, Hausner’s ratio, Carr’s index, angle of repose and bulk density have been considered to assess the flow potential of microcapsules of fish oil and antioxidants in recent studies. Hausner’s ratio and Carr’s index depend on bulk and tapped density (Hausner’s ratio = tapped density/bulk density; Carr’s index (%) = (tapped density − bulk density/tapped density)*100). The bulk density is determined by transferring the microcapsules into a graduated cylinder and recording the occupied volume. For the tapped density, the graduated cylinder is tapped from a height of about 5 cm until a constant volume is obtained and recorded. Then, the densities (g/mL) are calculated as the weight of the sample (g) divided by the corresponding volume (mL). The angle of repose (=tan − 1 (H/R)) is determined using a fixed funnel and a digital angle ruler to determine the height of the pile (H) and the radius at the base (R) [78]. Values of Hausner’s ratio, Carr’s index and angle of repose higher than 1.25, 25% and 45°, respectively, indicate poor flowability. Bulk and tapped density values lower than 0.30 and 0.45, respectively, are relevant for packaging and material handling purposes in the food industry [88]. Dadi et al. [78] have found the influence of the microencapsulation technique in these properties, with higher values of Hausner’s ratio, Carr’s index, and angle of repose in freeze-dried microcapsules in comparison to spray-dried ones, which the authors have related with the irregular shape of the former. In addition, in this study, no differences were observed in the flowing properties due to the wall material (maltodextrin vs. maltodextrin and high methoxyl pectin mixture). However, Bakry et al. [43] reported an increase in the values of Hausner’s ratio and Carr’s index in fish oil microcapsules to which carrageenan had been added, leading to more coherent and less competent microcapsules in flowing, and these results were attributed to the rise in the particle size and moisture content in these microcapsules.

When new vehicles of bioactive compounds are developed, it is necessary to know their gastrointestinal stability and bioavailability. However, few studies on fish oil and antioxidant microcapsules have carried out this type of analysis, consisting of a static and in vitro digestion that simulates gastric and intestinal phases. The release of bioactive compounds (antioxidants, fish oil, omega-3 fatty acids) at each stage is normally evaluated. A wide range of different conditions has been applied, making a meaningful comparison of results difficult. To avoid this difficulty in comparison, Minekus et al. [93] have recommended their standardized protocol that uses constant ratios of meal to digestive fluids and sequential oral, gastric and intestinal steps of digestion with constant pH and parameters such as electrolytes, enzymes, bile, dilution and time of digestion based on available physiological data. Microencapsulation techniques were found to have significantly influenced the releases of phenolic and flavonoic compounds from microcapsules of Moringa stenopetala leaf extract, with higher release from freeze-dried microcapsules, which can be explained by the lower microencapsulation efficiency [78]. The microencapsulation of fish oil using thiol-modified β-lactoglobulin fibril–chitosan complex may protect the core from gastric enzymes, as indicated by a lower fish oil release in the oral phase (<25%) and enhanced sustainable release of fish oil under intestinal conditions (reaching 70%) [33]. Solomando et al. [74] did not find differences between fish oil microcapsules of maltodextrin and maltodextrin–chitosan in the release of EPA and DHA during in vitro digestion of microcapsules, which principally took place during the first hour of the intestinal phase.

## 7. Food Enrichment

Although most recent studies on microencapsulation of fish oil and antioxidants have aimed to evaluate the emulsion constituents for microencapsulation and the procedure, some publications have also depicted the influence of adding the microencapsulates to some food products (Table 4). In the case of the fish oil microcapsules, Fadini et al. [41] prepared fish oil microcapsules, with skimmed milk powder, acacia gum and a mixture of acacia gum and grape juice (ratio of 1:1:1) and a second shell made of vegetable fat and hydrogenated palm oil, which were incorporated into melted chocolate to cover extruded cereals. In this study, the fish oil off-flavor was the only parameter evaluated in the enriched food, and it was evaluated by using trained assessors. They noted a slight difference in flavor between enriched and non-enriched samples, but the flavor in the enriched samples was not characterized as a fish oil flavor. The authors have ascribed this result to the grape juice of the wall material of the microcapsules, which may have had an influence on this perception. Similarly, in [44], a study considering reconstituted milk with the addition of zein fish oil microcapsules, a sensory analysis focused on overall fishiness attributes (taste, odor, flavor and appearance) was the only analysis carried out. These authors did not find significant differences between enriched and non-enriched samples. Fish oil microcapsules composed of soy protein isolate and inulin have been added to burgers, which were evaluated by means of pH, instrumental color and texture, fatty acid profile, volatile compounds and hedonic and CATA sensory analysis [72]. The findings of this study have shown a negative impact of the addition of fish oil microcapsules to the burgers, with a significant decrease in pH and overall liking and an increase in hardness, chewiness and volatile oxidation compounds. Consequently, the use of this type of fish oil microcapsules is not recommended as an approach for incorporating omega-3 fatty acids in meat products such as burgers. Other authors [75,76,77] have carried out an extensive investigation about the addition of fish oil microcapsules (with maltodextrin and maltodextrin–chitosan) to meat derivatives (cooked and dry-cured meat products). These authors have achieved cooked and dry-cured products that can be labeled as “source of omega-3 fatty acids” (at least 40 mg EPA + DHA/100 g sample), without influencing physicochemical characteristics; oxidative stability; acceptability; or usual changes that take place during the processing, storing or culinary heating of these products. The sensory properties of these omega-3-enriched meat products were widely evaluated by means of quantitative descriptive analysis, temporal dominance of sensations and hedonic and purchase intent; in general, there was a slight decrease in the score of flavor attributes and an increase in saltiness. The positive influence of the label information on hedonic and purchase intent was also noted. In addition, these authors have evaluated the bioaccessibility of EPA and DHA in the enriched meat products and found them to be higher when fish oil microcapsules with maltodextrin–chitosan were added. Thus, these authors have pointed out the viability of the use of these types of fish oil microcapsules to enrich meat products subjected to low and high temperatures for manufacturing, storage in refrigeration and culinary heating, recommending the adjustment of the content of salt and flavor additives and the inclusion of accurate label information. In another study, Solomando et al. [76] tested the addition of microcapsules of fish oil and fish oil–lycopene to dry-cured ham and cheese spreads. Results on proximal composition, oxidative stability, EPA and DHA enrichment and sensory analysis were quite satisfactory in the case of dry-cured ham spreads. However, the use of these types of microcapsules was not accurate in enriching cheese spreads. In this product, the quantity of EPA and DHA decreased during storage, and off-flavors were detected. 

Some recent studies have also evaluated the effect of adding antioxidant microcapsules to foods (Table 2). In the work of Hamid et al. [82], microcapsules of wild pomegranate flavedo phenolics were added to an ordinary ready-to-serve drink; they were found to be acceptable up to a microcapsule concentration of 2% (*w*/*w*) and resulted in significant improvement in phenolics, flavonoids and antioxidant activity. de Cássia Sousa Mendes et al. [83] have added microcapsules of jabuticaba extracts to biscuits, which became products with bioactive properties (with the presence of polyphenolic compounds and antioxidant potential). Gheonea (Dima) et al. [66] have used microcapsules of lycopene from tomato peels for functionalization in dressing samples, which showed increased antioxidant activity and typical solid-like behavior in the rheological tests.

## 8. Conclusions

The latest developments in the microencapsulation of fish oil and natural antioxidants have mainly evaluated the impact of the wall material and the procedure variables on the quality of the microencapsulates, while their addition to foods has been studied in few works. 

Three main types of polymers have been used for microencapsulation of fish oil and antioxidants: saccharides, which were the most used, followed by animal and vegetable proteins. Whey protein isolate or concentrate and maltodextrin have been predominantly used in most recent studies; however, their uses have been devoted to comparison and/or combination with less-tested materials, such as cellulose, inulin, fish protein isolate, chitosan, soy protein isolate, zein, alginates, starch, gelatin and sucrose. 

As for the procedure variables, studies have principally focused on the optimization of the homogenization technique of emulsion dissolution and the wall material ratio and the comparison of different microencapsulation techniques, with spray drying being the most used.

Few recent studies have evaluated the effect of adding fish oil and natural antioxidant microcapsules to food products, and these studies have been aimed at the influence on sensory quality and antioxidant properties, respectively.

The quality evaluation of the fish oil and natural antioxidant microcapsules has been assessed in most studies by determining size, microencapsulation efficiency, morphology, and moisture, as well as oxidation stability in the case of fish oil microcapsules and antioxidant properties, solubility and release for microcapsules of natural antioxidants. Nevertheless, valuable parameters such as flowing properties, yield percentage, bioaccessibility and biodisponibility have been used to a lesser extent. In addition, techniques widely used in other fields, such as NMR and FTIR, have been applied to evaluate some properties of microcapsules.

Thus, the advancements shown in recent studies on fish oil and natural antioxidant microcapsules have demonstrated the importance of evaluating the impact of using new materials or techniques on the quality of the obtained powders and optimizing the most influential variables of the procedure. Further studies should improve the selection of the quality analysis parameters for the microcapsules and consider the additional influence of the microcapsules on food products, paying special attention to the wall material that best matches each food matrix type. 

## Figures and Tables

**Figure 1 foods-11-03291-f001:**
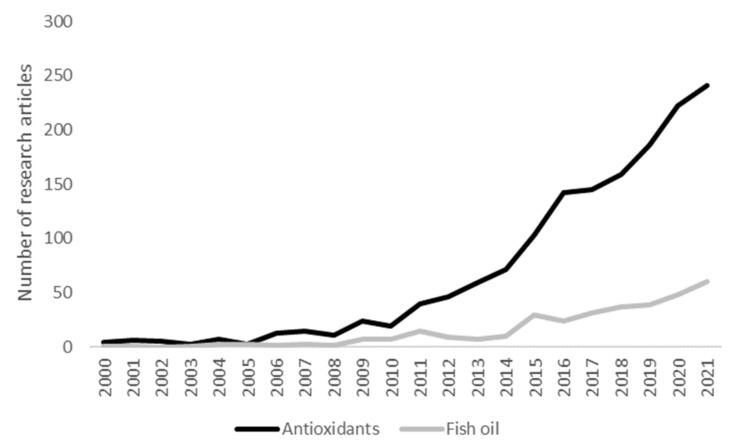
Chronological evolution in the number of research articles about microencapsulation of fish oil and natural antioxidants in the last two decades (data obtained from Web of Science, November 2021).

**Figure 2 foods-11-03291-f002:**
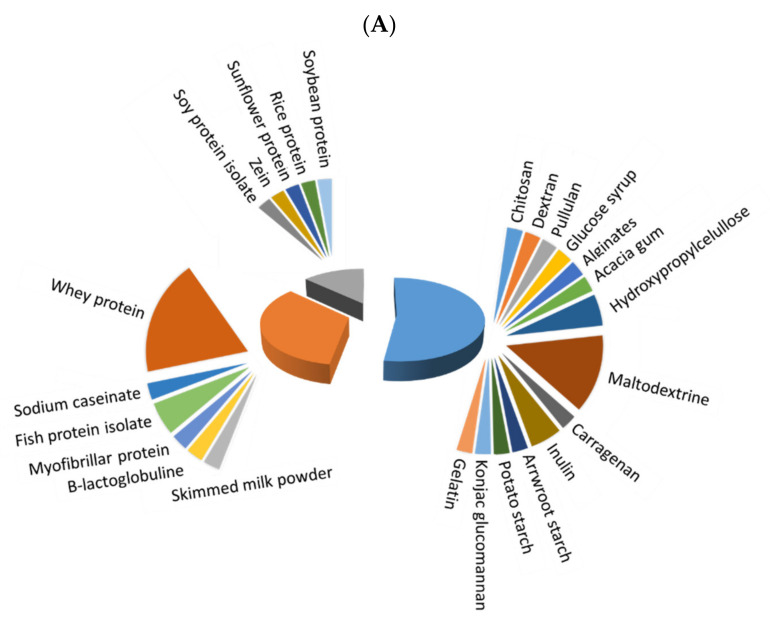
Extent of use of saccharides (blue), vegetable proteins (grey) and animal proteins (orange) as wall material for microencapsulation of fish oil (**A**) and natural antioxidants (**B**).

**Figure 3 foods-11-03291-f003:**
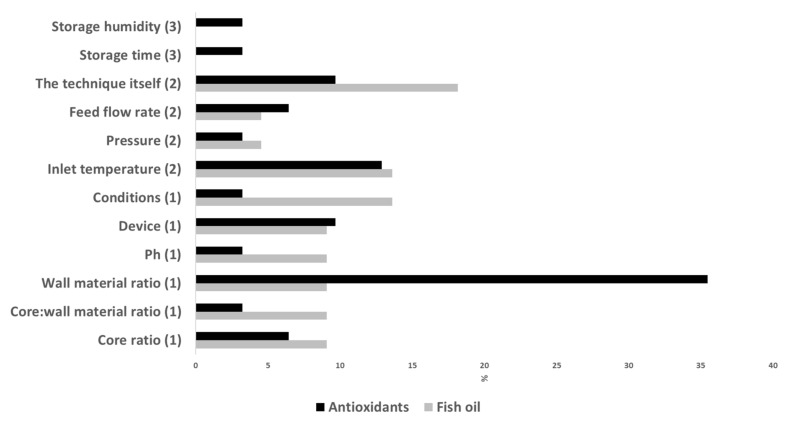
Extent of evaluation of different variables (related to the emulsion/solution preparation (1), the microencapsulation technique (2), and other parameters (3)) for the development of fish oil and natural antioxidant microcapsules.

**Figure 4 foods-11-03291-f004:**
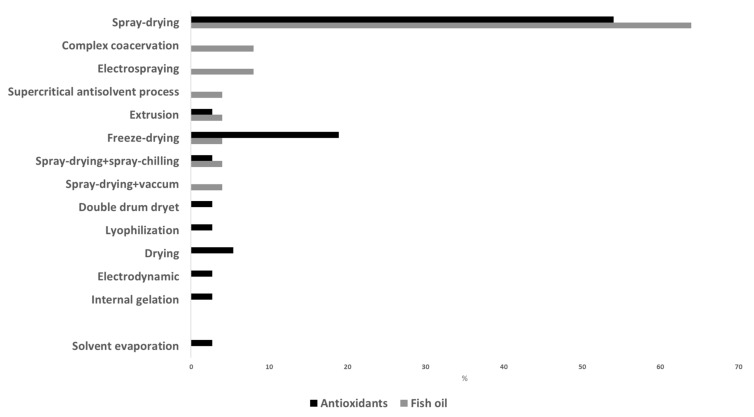
Extent of use of microencapsulation techniques for production of fish oil and antioxidant microcapsules.

**Figure 5 foods-11-03291-f005:**
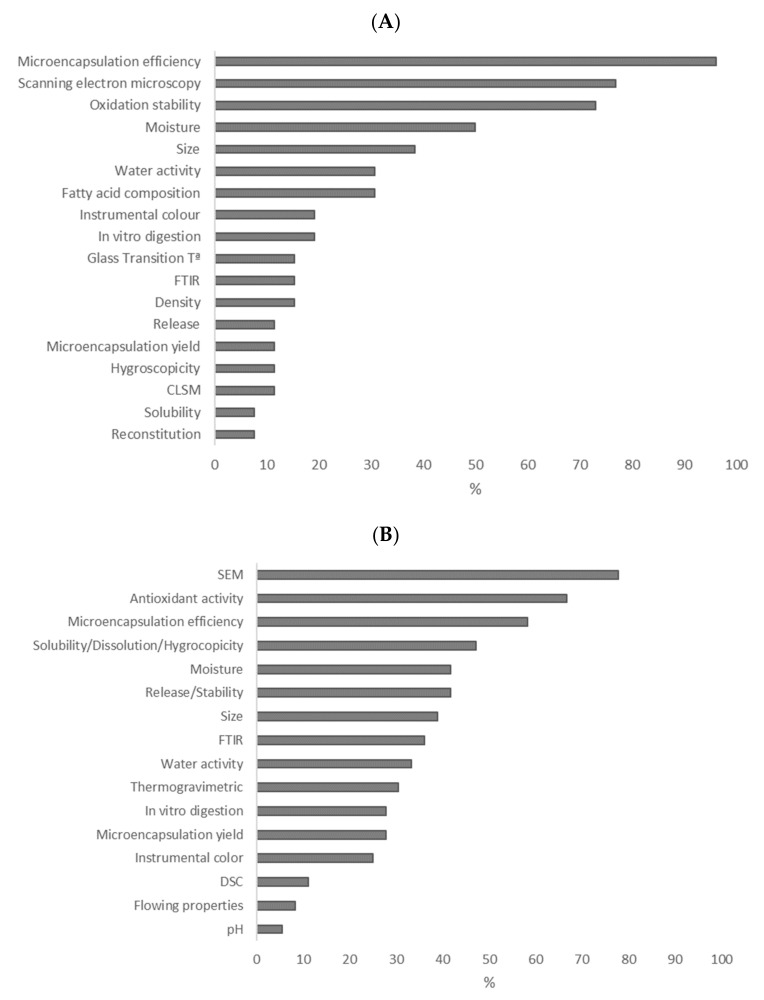
Extent of use of most applied analyses for the quality evaluation of fish oil microcapsules (**A**) and antioxidant microcapsules (**B**) in recent studies.

**Table 3 foods-11-03291-t003:** Most used techniques for the quality evaluation of fish oil and antioxidant microcapsules.

Technique	Conditions	Findings
Laser light diffraction	In dry dispersion or solvents	Size of microcapsules (high variability)
Solvent extraction	With petroleum ether or hexane	Microencapsulation efficiency (high variability)
Scanning electron microscopy		Morphology (visible wrinkles or dimples on the surface; spherical shape with dent; surface glassy and irregular shapes with edges; egg-shape morphology; broken microcapsules)
AOAC method (reference 935.29)	By drying	Moisture (<4%)
Thiobarbituric active reactive substance	Expressed as mg malondialdehyde/Kg microcapsule or fat	Secondary oxidation products
Gas chromatography–mass spectrometry detector	Coupled to head space microextraction Expressed as area units	Volatile compounds (no detection of compounds from omega-3 polyunsaturated fatty acids)
2,2-diphenyl-1-picrylhydrazyl (DPPH) free-radical scavenging method	Expressed as mg Trolox or ascorbic acid-equivalent per gram of microencapsulate	Antioxidant activity
Attenuated total reflection Fourier transform infrared	Based on the intensity of the characteristic absorption band of omega-3 PUFAs at 3012 cm^−1^	Oxidation stability
Proton nuclear magnetic resonance	Relative intensities of key peaks associated with the polyunsaturated alkyl chain	Oxidation stability
Dissolving–centrifuging–drying	Expressed as the percentage of dissolution or the time that it takes for the powder to completely dissolve	Solubility
Fourier transform infrared	Characteristic bands	Interactions between wall material and core
Differential scanning calorimetry	Glass transition temperature is measured	Physical behavior of materials
Use of a graduated cylinder	Recording the volume	Bulk density; tapped density; Hausner’s ratio; Carr’s index
Static and in vitro digestion	A standardized protocol is recommended	Gastrointestinal stability and bioavailability

**Table 4 foods-11-03291-t004:** Enrichment of foods with fish oil and natural antioxidant microcapsules: main details.

Microcapsules	Enriched Food	Analysis	Results
Fish Oil			
Wall 1: skimmed milk powder, acacia gum, and grape juice (1:1:1) Wall 2: vegetable fat and hydrogenated palm oil Core: fish oil	Melted chocolate	Off-flavor	➢ Trained assessors noted a slight difference in flavor between enriched and non-enriched samples, but the flavor in enriched samples was not characterized as a fish oil flavor
Wall: zein Core: fish oil	Reconstituted milk	Sensory analysis: taste, odor, flavor and appearance.	➢ No significant differences between enriched and non-enriched samples
Wall: soy protein isolate and inulin Core: fish oil	Burgers	pH Instrumental color Instrumental texture Fatty acid profile Volatile compounds Sensory analysis: hedonic and CATA	➢ Significant decrease in pH ➢ Significant decrease in overall liking ➢ Significant increase in hardness and chewiness ➢ Significant increase in volatile oxidation compounds ➢ Negative impact of the fish oil microcapsule addition
Wall: maltodextrin or maltodextrin–chitosan Core: fish oil	Cooked and dry-cured sausages	Quantity of fatty acids Lipid oxidation Sensory analysis Bioaccessibility Volatile compounds Sensory analysis: hedonic and CATA	➢ >40 mg EPA + DHA/100 g sample ➢ High lipid oxidation stability ➢ Significant increase in saltiness ➢ High scores for acceptability and purchase intent with label information ➢ Higher bioaccessibility with maltodextrin–chitosan microcapsules
Wall: maltodextrin–chitosan Core: fish oil + lycopene	Spreads	Proximal composition Quantity of fatty acids Lipid oxidation Sensory analysis	➢ Quite satisfactory in the case of dry-cured ham spreads ➢ Unsatisfactory results in cheese spreads
Natural Antioxidants			
Wall: maltodextrin Core: pomegranate flavedo extract	Fruit drink	Content in phenolics Content in flavonoids Antioxidant activity Acceptability	➢ Improvement in phenolics, flavonoids and antioxidant activity ➢ Good acceptability up to the microcapsule concentration of 2% (*w*/*w*)
Wall: sodium alginate Core: jabuticaba extracts	Cassava starch biscuits	Content in phenolics Antioxidant activity	➢ Moderate losses in phenolics and antioxidant activity during baking biscuits
Wall: whey protein isolate and acacia gum Core: tomato peel extract	Dressings	Antioxidant activity Rheological test	➢ Increased antioxidant activity ➢ Typical solid-like

## Data Availability

Data is contained within the article.

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
