# Peer review of "Recent Developments in the Microencapsulation of Fish Oil and Natural Extracts: Procedure, Quality Evaluation and Food Enrichment"

_foods, 2022, doi:10.3390/foods11203291_

Round 1
Reviewer 1 Report
Since the paper did not present the antioxidant activity of the extracts and the influence of microencapsulation on the antioxidant properties, the title should be changed to „...natural extracts...“ instead of „....natural antioxidants...“
Fig 2 should be presented in more readable manner.
Line 111-123 the data is repeated from the lines 89-101
Line 251 bracket is in excess
Fig 3 Font should be larger, it is hard to read
Numeration of the pages was not performed well
Lines 441-462 too broad explanation of supercritical fluids was given with only one application. Please give more application of this process in fish oil encapsulation
464-478 No application of ionotropic gelation method or simple solvent evaporation is presented for fish oil
Line 533 remove the year of publication
140 references were cited in the text but only 96 in the reference list
Author Response
We are very thankful to the reviewer 1 for their interest in improving the quality of this manuscript by providing sensible comments. Their reports and recommendations are, undoubtedly, very valuable. We hope this revision fulfils your expectations. The responses to the reviewer inquiries and the modifications included in the new manuscript are written in red and listed below, one by one (the specific pages and lines correspond to the present version of the manuscript):
R1_1. Since the paper did not present the antioxidant activity of the extracts and the influence of microencapsulation on the antioxidant properties, the title should be changed to „...natural extracts...“ instead of „....natural antioxidants...“
As reviewer indicated, ´extracts´ instead of ´antioxidants´ in title.
R1_2. Fig 2 should be presented in more readable manner
According to reviewer´s comment, authors have tried to improve fig. 2.
R1_3. Line 111-123 the data is repeated from the lines 89-101
That was a mistake, which has been corrected by eliminating lines 111-123.
R1_4. Line 251 bracket is in excess
A bracket has been eliminated in line 262 of the present version of the manuscript, as reviewer indicated.
R1_5. Fig 3 Font should be larger, it is hard to read.
Font of fig. 3 has been changed according to reviewer´s comment.
R1_6. Numeration of the pages was not performed well
Since the manuscript is prepared following a template provided by the journal, authors can not modify the numeration of the pages.
R1_7. Lines 441-462 too broad explanation of supercritical fluids was given with only one application. Please give more application of this process in fish oil encapsulation
Authors have described, for each technique, those publications found according to the search criteria, and more applications were not found for supercritical fluids.
R1_8. 464-478 No application of ionotropic gelation method or simple solvent evaporation is presented for fish oil.
Authors have described, for each technique, those publications found according to the search criteria, and no applications of ionotropic gelation method or simple solvent evaporation were found for fish oil.
R1_9. Line 533 remove the year of publication
That was a mistake, which has been corrected by eliminating ´(2021)´ in line 526 of the present version of the manuscript.
R1_10. 140 references were cited in the text but only 96 in the reference list
There were some mistakes that has been amended.
Reviewer 2 Report
Dear authors, The review article submitted for review is very interesting. The individual chapters clearly lead over from one aspect to another. There are no basic messages used in the subsections duplicating other articles, which I like. I feel that the final section - the conclusion - needs more development and attention. What I miss here is the authors' own opinion on the subject under discussion, what they foresee as the next steps in the development of this technique, what they think would be interesting.
Author Response
We are very thankful to the reviewer 2 for their interest in improving the quality of this manuscript by providing sensible comments. Their reports and recommendations are, undoubtedly, very valuable. We hope this revision fulfils your expectations. The responses to the reviewer inquiries and the modifications included in the new manuscript are written in red and listed below, one by one (the specific pages and lines correspond to the present version of the manuscript):
R2_1. Dear authors, the review article submitted for review is very interesting. The individual chapters clearly lead over from one aspect to another. There are no basic messages used in the subsections duplicating other articles, which I like.
Authors are glad about this comment.
R2_2. I feel that the final section - the conclusion - needs more development and attention. What I miss here is the authors' own opinion on the subject under discussion, what they foresee as the next steps in the development of this technique, what they think would be interesting.
Last sentences of the conclusion showed authors` opinion on further studies, which has been extended in the present version of the manuscript (lines 785-788).
Reviewer 3 Report
I reviewed the manuscript, entitled, Recent developments in the microencapsulation of fish oil and natural antioxidants: procedure, quality evaluation and food enrichment. The manuscript has a lot of flaws and the authors should consider below suggestions to improve.
Title: please remove the dot after the title
Introduction
How did authors obtain Figure 1? It seems like WoS. Please mention this.
Figure 1, Y axis, put the line and is it only a research article? Or review. Book chapters, conference proceedings?
Section 2: please add the date and month of search operation. When was the last update of WoS? This should be added to avoid bias
Figure 2 should be improved. It is not readable text
Constituents of emulsions/solutions for microencapsulation: Authors should separate plant and animal-based wall materials and should discuss recent trends. What is the progress of emulsion/ solutions in microencapsulation?
Figures 3 and 4 quality is extremely low.
Procedure for microencapsulation; what are the recent trends and advances in microencapsulation should be introduced in a Table. Author simply presented the old techniques. This review should contribute to understanding the microencapsulation advances.
Microcapsules quality evaluation: Authors should formulate table containing different techniques used and their findings; conditions of the techniques used
Food enrichment: Please add Table containing microcapsules; fortified product (bread or ice cream, milk); what is the advantage; storage studies, if conducted; stability studies;
References are nor according to the journal format
Author Response
We are very thankful to the reviewer 3 for their interest in improving the quality of this manuscript by providing sensible comments. Their reports and recommendations are, undoubtedly, very valuable. We hope this revision fulfils your expectations. The responses to the reviewer inquiries and the modifications included in the new manuscript are written in red and listed below, one by one (the specific pages and lines correspond to the present version of the manuscript):
R3_1. I reviewed the manuscript, entitled, Recent developments in the microencapsulation of fish oil and natural antioxidants: procedure, quality evaluation and food enrichment. The manuscript has a lot of flaws and the authors should consider below suggestions to improve.
Thanks a lot.
R3_2. Title: please remove the dot after the title
As reviewer indicated, ´.´ has been removed after the title.
R3_3. Introduction
How did authors obtain Figure 1? It seems like WoS. Please mention this.
Figure 1, Y axis, put the line and is it only a research article? Or review. Book chapters, conference proceedings?
As indicated, in Y axis and caption of fig. 1, it refers to research articles. This figure has been created by authors with obtained from search in WoS in November 2021, which has been indicated in caption of fig.1 the present version of the manuscript (lines 73-74).
R3_4. Section 2: please add the date and month of search operation. When was the last update of WoS? This should be added to avoid bias.
This information has been included in the present version of the manuscript (line 79).
R3_5. Figure 2 should be improved. It is not readable text
According to reviewer´s comment, authors have tried to improve fig. 2.
R3_6. Constituents of emulsions/solutions for microencapsulation: Authors should separate plant and animal-based wall materials and should discuss recent trends. What is the progress of emulsion/ solutions in microencapsulation?
According to the reviewer´s comment about plant and animal-based wall materials, a sentence has been included in section 3 (lines 103-105). Regarding the progress of emulsions/solutions in microencapsulation, it is indicated in the conclusion section (Whey protein isolated or concentrated and maltodextrin have been predominantly used in most recent studies, however, its uses has been devoted to being compared and/or combined with less experimented materials, such as cellulose, inulin, fish protein isolate, chitosan, soy protein isolate, zein, alginates, starch, gelatine, or sucrose).
R3_7. Figures 3 and 4 quality is extremely low.
According to reviewer´s comment, authors have tried to improve fig. 3 and 4.
R3_8. Procedure for microencapsulation; what are the recent trends and advances in microencapsulation should be introduced in a Table. Author simply presented the old techniques. This review should contribute to understanding the microencapsulation advances.
As explained in the manuscript, this work aims to review most recent developments in the microencapsulation of fish oil and natural antioxidant compounds to fortify foods, focusing on their key objectives and the quality evaluation of the microcapsules. Thus, we have not aimed to search about microencapsulation techniques. In fact, there some review articles focused on this aspect.
R3_9. Microcapsules quality evaluation: Authors should formulate table containing different techniques used and their findings; conditions of the techniques used.
Authors think that specifying the conditions of each technique and the obtained results of the reviewed papers does not make sense; it would be to transcribe the cited articles. As the article are referenced, readers can access them for details.
R3_10. Food enrichment: Please add Table containing microcapsules; fortified product (bread or ice cream, milk); what is the advantage; storage studies, if conducted; stability studies;
Authors have not considered to make a specific table for food enrichment because there is not much data about this issue. It has been included in table 1 and 2, for fish oil and natural antioxidants, respectively, indicating the fortified product, the core mare material of the microcapsules and the microencapsulation techniques. Details of each article have been summarized in the text.
R3_11. References are nor according to the journal format.
There were some mistakes that has been amended.
Round 2
Reviewer 3 Report
I re-reviewed the manuscript entitled, Recent developments in the microencapsulation of fish oil and natural antioxidants: procedure, quality evaluation and food enrichment. Authors failed to address the many suggestions made by me. In my opinion, without the addition of recent trends in microencapsulation of techniques, it won’t contribute to the field.
Unanswered/avoided questions
Constituents of emulsions/solutions for microencapsulation: Authors should separate plant and animal-based wall materials and should discuss recent trends. What is the progress of emulsion/ solutions in microencapsulation?
Authors did not include the trends in this section.
Procedure for microencapsulation; what are the recent trends and advances in microencapsulation should be introduced in a Table. Author simply presented the old techniques. This review should contribute to understanding the microencapsulation advances.
I suggested authors include recent trends in microencapsulation of fish oil. In this section, authors simply presented very old techniques for encapsulation of fish or natural antioxidants. Based on my experience, there are many new techniques available to encapsulate fish or natural antioxidants.
Microcapsules quality evaluation: Authors should formulate Table containing different techniques used and their findings; conditions of the techniques used.
I suggested authors to include Table containing quality evaluation; however, authors ignore to include it
Food enrichment: Please add Table containing microcapsules; fortified product (bread or ice cream, milk); what is the advantage; storage studies, if conducted; stability studies;
There are many studies available in the literature on fortification in food products, such as bread, ice cream.
All suggested comments are feasible to include in the review paper, however, authors ignored the suggestions and made no changes.
Author Response
We are very thankful to the reviewer 3 for their interest in improving the quality of this manuscript by providing sensible comments. Their reports and recommendations are, undoubtedly, very valuable. We hope this second revision fulfils your expectations. The responses to the reviewer inquiries and the modifications included in the new manuscript are written in blue and listed below, one by one (the specific pages and lines correspond to the present version of the manuscript):
R3_1. Constituents of emulsions/solutions for microencapsulation: Authors should separate plant and animal-based wall materials and should discuss recent trends. What is the progress of emulsion/ solutions in microencapsulation? Authors did not include the trends in this section.
According to the reviewer´s comment, three paragraphs have been included in the present version of the manuscript: lines 104-180, 127-130 and 231-234.
R3_2. Procedure for microencapsulation; what are the recent trends and advances in microencapsulation should be introduced in a Table. Author simply presented the old techniques. This review should contribute to understanding the microencapsulation advances. I suggested authors include recent trends in microencapsulation of fish oil. In this section, authors simply presented very old techniques for encapsulation of fish or natural antioxidants. Based on my experience, there are many new techniques available to encapsulate fish or natural antioxidants.
Authors have presented the most used microencapsulation techniques in the papers that have been selected for this review according to the criteria search. Besides, those articles that have focused on evaluating the microencapsulation procedure have been presented in table 2 and 3 (in the procedure section). And the trend followed in these recent selected articles have been summarized in section 5, including figures 3 and 4.
R3_3. Microcapsules quality evaluation: Authors should formulate Table containing different techniques used and their findings; conditions of the techniques used.
According to the reviewer´s comment, a new table (Table 3) has been included in the present version of the manuscript. This new table has also been introduced in the manuscript (lines 525-526).
R3_4. Food enrichment: Please add Table containing microcapsules; fortified product (bread or ice cream, milk); what is the advantage; storage studies, if conducted; stability studies; There are many studies available in the literature on fortification in food products, such as bread, ice cream.
According to the reviewer´s comment, a new table (Table 4) has been included in the present version of the manuscript. This new table has also been introduced in the manuscript (line 731). We know that there are more studies than those cited in the present revision in the scientific literature, however, authors have selected for this review those that fit the criteria search.